# Determination of the Binding Epitope of an Anti-Mouse CCR9 Monoclonal Antibody (C_9_Mab-24) Using the 1× Alanine and 2× Alanine-Substitution Method

**DOI:** 10.3390/antib12010011

**Published:** 2023-01-31

**Authors:** Hiyori Kobayashi, Teizo Asano, Tomohiro Tanaka, Hiroyuki Suzuki, Mika K. Kaneko, Yukinari Kato

**Affiliations:** 1Department of Molecular Pharmacology, Tohoku University Graduate School of Medicine, 2-1 Seiryo-machi, Aoba-ku, Sendai 980-8575, Japan; 2Department of Antibody Drug Development, Tohoku University Graduate School of Medicine, 2-1 Seiryo-machi, Aoba-ku, Sendai 980-8575, Japan

**Keywords:** mouse CCR9, monoclonal antibody, epitope mapping, alanine scanning, enzyme-linked immunosorbent assay

## Abstract

C-C chemokine receptor 9 (CCR9) is a receptor for C-C-chemokine ligand 25 (CCL25). CCR9 is crucial in the chemotaxis of immune cells and inflammatory responses. Moreover, CCR9 is highly expressed in tumors, including several solid tumors and T-cell acute lymphoblastic leukemia. Several preclinical studies have shown that anti-CCR9 monoclonal antibodies (mAbs) exert antitumor activity. Therefore, CCR9 is an attractive target for tumor therapy. In this study, we conducted the epitope mapping of an anti-mouse CCR9 (mCCR9) mAb, C_9_Mab-24 (rat IgG_2a_, kappa), using the 1× alanine (1× Ala)- and 2× alanine (2× Ala)-substitution methods via enzyme-linked immunosorbent assay. We first performed the 1× Ala-substitution method using one alanine-substituted peptides of the mCCR9 N-terminus (amino acids 1–19). C_9_Mab-24 did not recognize two peptides (F14A and F17A), indicating that Phe14 and Phe17 are critical for C_9_Mab-24-binding to mCCR9. Furthermore, we conducted the 2× Ala-substitution method using two consecutive alanine-substituted peptides of the mCCR9 N-terminus, and showed that C_9_Mab-24 did not react with four peptides (M13A–F14A, F14A–D15A, D16A–F17A, and F17A–S18A), indicating that _13-_MFDDFS_-18_ is involved in C_9_Mab-24-binding to mCCR9. Overall, combining, the 1× Ala- or 2× Ala-scanning methods could be useful for understanding for target–antibody interaction.

## 1. Introduction

C-C chemokine receptor 9 (CCR9) is a member of the G-protein coupled receptors with seven transmembrane domains and four extracellular regions. Previous studies have shown that CCR9 is expressed on the surface of immature T lymphocytes and intestinal cells [1,2]. The C-C chemokine ligand 25 (CCL25), the only ligand for CCR9 [3,4], is mainly expressed in the thymus and intestinal epithelium [3,5,6,7]. CCL25 induces the chemotaxis of immature T cells into the thymus for their maturation [8]. Studies have demonstrated that CCR9 and CCL25 play important roles in inflammatory diseases, such as asthma [9], inflammatory bowel disease [10,11,12], and hepatitis [13,14]. Moreover, CCR9 is highly expressed in malignancies, such as lung cancer [15], breast cancer [16,17], ovarian cancer [18], melanoma [19,20], and T-cell acute lymphoblastic leukemia (T-ALL) [21]. CCR9 is expressed in more than 70% cases of T-ALL. However, it is expressed in a small subset of normal T cells [21]. The interaction of CCR9 and CCL25 activates the phosphatidylinositide 3-kinase (PI3K)/Akt signaling pathway, which is involved in the proliferation and survival of tumor cells [18,22,23].

Several anti-CCR9 mAbs have been developed for therapeutic uses. Anti-human CCR9 mAbs (clones 91R and 92R) exhibited cytotoxicity against CCR9-positive tumors via antibody-dependent cell-mediated cytotoxicity and complement-dependent cytotoxicity. These antibodies suppressed T-ALL proliferation in mouse xenograft models [24,25]. Moreover, Maciocia et al. developed a CCR9-specific mAb by gene-gun vaccination of rats with a plasmid encoding human CCR9. They further produced the chimeric antigen receptor (CAR)-T cells, and showed a potent antitumor effect in cell lines and patient-derived xenograft mouse models of T-ALL [21]. Therefore, CCR9 is considered as an attractive target for tumor therapy [5,26].

Using the N-terminal peptide immunization, we developed anti-mouse chemokine receptor mAbs against CCR2 [27], CCR3 [28], CCR4 [29], CCR6 [30], and CXCR6 [31]. Furthermore, we developed an anti-mouse CCR9 (mCCR9) mAb, C_9_Mab-24 (rat IgG_2a_, kappa), via peptide immunization of the mCCR9 N-terminus (amino acids 1–19) [32]. C_9_Mab-24 could be applied to flow cytometry in mCCR9-overexpressed Chinese hamster ovary-K1 cells and endogenously expressed RL2 cells [32]. In this study, we determined the binding epitope of C_9_Mab-24 using two different alanine scanning strategies via enzyme-linked immunosorbent assay (ELISA).

## 2. Materials and Methods

### 2.1. Development of C_9_Mab-24

Eurofins Genomics KK (Tokyo, Japan) synthesized a partial sequence of the N-terminal extracellular region of mCCR9 (Accession No.: NP_001160097) with cysteine at its C-terminus (mCCR9p1-19C; sequence: MMPTELTSLIPGMFDDFSYC). Subsequently, the keyhole limpet hemocyanin (KLH) was conjugated at the C-terminus of the peptide (mCCR9p1-19C-KLH). Chinese hamster ovary (CHO)-K1 cells were obtained from the American Type Culture Collection (Manassas, VA, USA). The expression plasmid of mCCR9 (pCMV6neo-mCCR9-Myc-DDK) was purchased from OriGene Technologies, Inc. (Rockville, MD, USA). The mCCR9 plasmid was transfected into CHO-K1 cells, using a neon transfection system (Thermo Fisher Scientific Inc., Waltham, MA, USA). Stable transfectants were established through cell sorting using a cell sorter (SH800; Sony Corp., Tokyo, Japan) using Roswell Park Memorial Institute (RPMI) 1640 medium (Nacalai Tesque, Inc.), containing 10% heat-inactivated fetal bovine serum (FBS, Thermo Fisher Scientific Inc.), 100 units/mL of penicillin, 100 μg/mL of streptomycin, and 0.25 μg/mL of amphotericin B (Nacalai Tesque, Inc.), and 0.5 mg/mL of G418 (Nacalai Tesque, Inc., Kyoto, Japan). CHO-K1, P3U1, mCCR9-overexpressed CHO-K1 (CHO/mCCR9), and RL2 were cultured in RPMI 1640 medium with 10% heat-inactivated FBS, 100 units/mL of penicillin, 100 μg/mL of streptomycin, and 0.25 μg/mL of amphotericin B. Cells were grown in a humidified incubator at 37 °C, at an atmosphere of 5% CO_2_ and 95% air.

A five-week old Sprague–Dawley rat was purchased from CLEA Japan (Tokyo, Japan). Animals were housed under specific pathogen-free conditions. All animal experiments were also conducted according to relevant guidelines and regulations to minimize animal suffering and distress in the laboratory. The Animal Care and Use Committee of Tohoku University (Permit number: 2019NiA-001) approved the animal experiments. The rat was monitored daily for health during the full four-week duration of the experiment. A reduction of more than 25% of the total body weight was defined as a humane endpoint. During sacrifice, the rat was euthanized through cervical dislocation, after which death was verified through respiratory and cardiac arrest. To develop mAbs against mCCR9, one rat was immunized intraperitoneally, using 100 µg mCCR9p1-19C-KLH peptide with Imject Alum (Thermo Fisher Scientific Inc.). The procedure included three additional immunizations, which were followed by a final booster intraperitoneal injection, two days before the harvest of spleen cells. Harvested spleen cells were subsequently fused with P3U1 cells, using PEG1500 (Roche Diagnostics, Indianapolis, IN, USA), after which hybridomas were grown in an RPMI medium supplemented with hypoxanthine, aminopterin, and thymidine for the selection (Thermo Fisher Scientific Inc.). Supernatants were subsequently screened with the mCCR9p1-19C peptide, using ELISA, following flow cytometry, using CHO/mCCR9, CHO-K1 and RL2 cells.

### 2.2. ELISA

The mCCR9 peptides, such as wild type (WT), 19 of 1× alanine (1× Ala)-substituted peptides (Table 1), and 18 of 2× alanine (2× Ala)-substituted peptides (Table 2), were synthesized using PEPscreen (Sigma-Aldrich Corp., St. Louis, MO, USA). Each peptide was immobilized on Nunc Maxisorp 96-well immuno plates (Thermo Fisher Scientific, Inc.) at a concentration of 1 μg/mL for 30 min at 37 °C. As a negative control, no peptide was immobilized on the immuno plates. After washing with phosphate-buffered saline containing 0.05% Tween20 (PBST), the wells were blocked with 1% bovine serum albumin containing PBST for 30 min at 37 °C. The plates were then incubated with C_9_Mab-24 (1 μg/mL), followed by a 1:20,000 dilution of peroxidase-conjugated anti-rat immunoglobulins (Sigma-Aldrich Corp.). Enzymatic reactions were performed using an ELISA POD Substrate TMB Kit (Nacalai Tesque, Inc.). Optical density was detected at 655 nm using an iMark microplate reader (Bio-Rad Laboratories, Inc., Berkeley, CA, USA).

## 3. Results

### 3.1. Epitope Determination Using 1× Ala-substituted mCCR9 Peptides

We previously developed an anti-mCCR9 mAb, C_9_Mab-24 (rat IgG_2a_, kappa) using peptide immunization of the mCCR9 N-terminus (amino acids 1–19) as shown in Materials and Methods [32]. C_9_Mab-24 is very useful for flow cytometry. Herein, we determined the binding epitope of C_9_Mab-24 using two different alanine scanning strategies via ELISA.

We synthesized 19 different 1× Ala-substituted mCCR9 peptides (Table 1). According to the ELISA results, C_9_Mab-24 reacted to 1× Ala-substituted peptides, such as M1A, M2A, P3A, T4A, E5A, L6A, T7A, S8A, L9A, I10A, P11A, G12A, M13A, D15A, D16A, S18A, and Y19A, as well as WT (positive control) (Figure 1A). In contrast, C_9_Mab-24 did not react with 1× Ala-substituted peptides, such as F14A and F17A as well as a negative control (Figure 1A). These results indicate that Phe14 and Phe17 of mCCR9 are the critical residues for the C_9_Mab-24 binding to mCCR9. Figure 1B summarizes the results. An alignment of mouse, rat, and human CCR9 sequences (residues, 1–19) is shown in Figure 1C.

### 3.2. Epitope Determination Using 2× Ala-substituted mCCR9 Peptides

The result of 1× Ala substitution showed that Phe14 and Phe17 of mCCR9 are the most critical for the C_9_Mab-24-mCCR9 interaction, but did not show that only Phe14 and Phe17 of mCCR9 are enough for C_9_Mab-24 binding to mCCR9. Therefore, we further investigated the C_9_Mab-24 epitope using 2× Ala-substituted mCCR9 peptides, as described previously [33].

We synthesized 18 different 2× Ala-substituted mCCR9 peptides (Table 2). According to the ELISA results, C_9_Mab-24 reacted to 2× Ala-substituted peptides, such as M1A–M2A, M2A–P3A, P3A–T4A, T4A–E5A, E5A–L6A, L6A–T7A, T7A–S8A, S8A–L9A, L9A–I10A, I10A–P11A, P11A–G12A, G12A–M13A, and S18A–Y19A as well as WT (positive control) (Figure 2A). In contrast, C_9_Mab-24 did not react with 2× Ala-substituted peptides, such as M13A–F14A, F14A–D15A, D16A–F17A, and F17A–S18A along with the negative control (Figure 2A). Moreover, the reactivity of C_9_Mab-24 with D15A–D16A was weaker than that with WT (Figure 2A). These results indicate that _13-_MFDDFS_-18_ is involved in C_9_Mab-24-binding to mCCR9. Figure 2B summarizes the results.

## 4. Discussion

The alanine-scanning mutagenesis method was first applied to antibody–antigen interaction in the human growth hormone (hGH) and 21 different anti-hGH mAbs using ELISA [34]. Single alanine mutations were introduced at every residue within the side chains that have been suggested in mAbs recognition. Using the method, the high-resolution “functional epitopes” could be mapped for each mAb [34].

We have established various anti-chemokine receptor mAbs against mouse CCR3 [35,36], mouse CCR8 [37,38,39], and human CCR9 (hCCR9) (clone C_9_Mab-1) [40] using the Cell-Based Immunization and Screening (CBIS) method. We also identified the epitope of C_9_Mab-1 in the N-terminal region of hCCR9 [41]. Then, we immunized the peptide of the region, and established a clone, C_9_Mab-11, which possesses a binding affinity comparable to C_9_Mab-1 [42]. C_9_Mab-1 possesses a wider epitope (_10-_IPNMA_-14_ and _16-_DY_-17_) [41] than that of C_9_Mab-11 (_11-_PNMA_-14_) (manuscript submitted). In this study, we identified the epitope of the anti-mCCR9 mAb, C_9_Mab-24, which was established by the N-terminal peptide immunization of mCCR9 [32]. Using the 2× Ala- and 1× Ala-substitution methods, we found that _13-_MFDDFS_-18_ is involved in C_9_Mab-24-binding to mCCR9, and the phenylalanines at 14 and 17 are critical among the amino acids (Figure 1 and Figure 2). The combination of 2× Ala- and 1× Ala-substitution methods could contribute the determination of the region and center of the mAb epitope.

We previously applied the strategy to identify the epitope of an anti-mouse CXCR6 mAb, Cx_6_Mab-1 [33]. First, we could not identify the epitope of Cx_6_Mab-1 by 1× Ala-substitution methods. Next, we performed the 2× Ala-substitution method, and could determine the epitope of Cx_6_Mab-1. The 2× Ala substitution could be another option to determine the epitope of mAbs if the epitope was not determined by the 1× Ala-substitution method.

The N-terminal region of CCR9 is important for the CCL25 interaction. Anti-hCCR9 mAbs (91R and 92R) recognized the N-terminus of hCCR9, which was partially inhibited in the presence of CCL25 [3,25]. The epitopes of 91R and 92R are located within 11–16 amino acids of hCCR9 [24,25], which is almost identical to the epitopes of C_9_Mab-1 and C_9_Mab-11. C_9_Mab-24 also binds to the N-terminal region of mCCR9 (_13-_MFDDFS_-18_). Therefore, both the neutralizing and biological activities of C_9_Mab-24 should be investigated in the future.

The therapeutic success for refractory childhood leukemia relies on the development of CAR-T targeting B-cell-specific antigen CD19 for B-ALL [43]. Although the therapy targets the common B cell antigen, the background of the success is the ability to tolerate B-cell aplasia. Compared to B-cell aplasia, T-cell aplasia exhibits intolerable immunosuppression. Therefore, a major hurdle for the development of CAR-T cell therapy for T-ALL is the inability to find T-ALL antigens that are not expressed in normal T-cells [44]. To overcome this problem, CCR9 is expected for cell surface antigens unique to T-ALL cells. The CAR-CCR9, a CAR-T with a single-chain variable fragment (scFv) for hCCR9, demonstrated cytotoxicity against CCR9-positive but not CCR9-negative T-ALL cells [21]. Although the epitope of the scFv has not been reported, the investigation of a suitable epitope to exert the CAR-T-mediated cytotoxicity is thought to be important for the future therapeutic applications of our anti-CCR9 mAbs.

## Figures and Tables

**Figure 1 antibodies-12-00011-f001:**
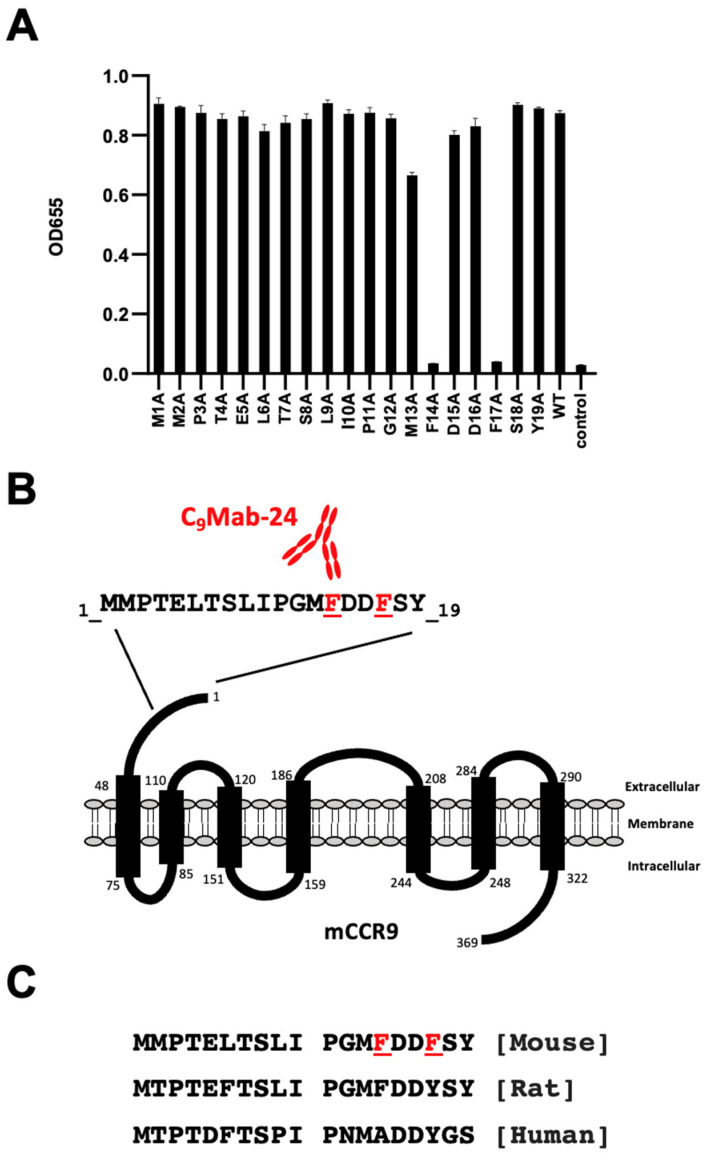
Determination of the C_9_Mab-24 epitope for mCCR9 by ELISA using 1× Ala-substituted peptides. (**A**) Synthesized 1× Ala-substituted peptides of mCCR9 were immobilized on immuno plates. The plates were incubated with C_9_Mab-24 (1 μg/mL), followed by peroxidase-conjugated anti-rat immunoglobulins. (**B**) Schematic illustration of mCCR9 and the critical amino acids for C_9_Mab-24 epitope. (**C**) An alignment of mouse, rat, and human CCR9 sequences (residues, 1–19).

**Figure 2 antibodies-12-00011-f002:**
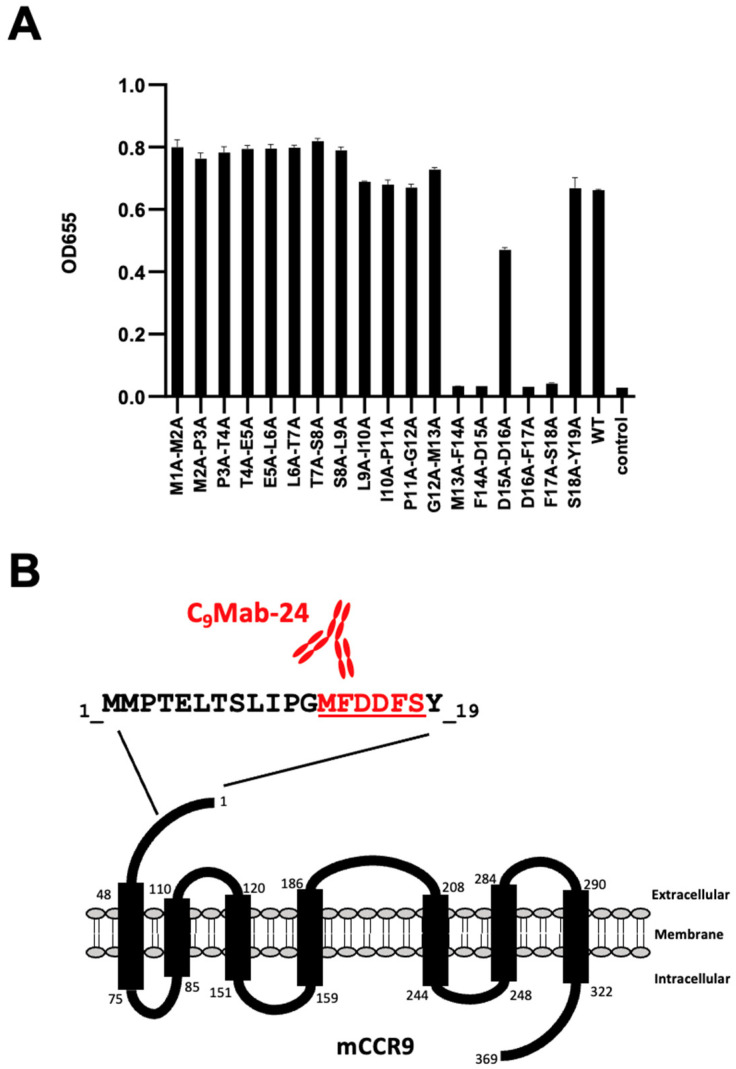
Determination of the C_9_Mab-24 epitope for mCCR9 by ELISA using 2× Ala-substituted peptides. (**A**) Synthesized 2× Ala-substituted peptides of mCCR9 were immobilized on immuno plates. The plates were incubated with C_9_Mab-24 (1 μg/mL), followed by peroxidase-conjugated anti-rat immunoglobulins. (**B**) Schematic illustration of mCCR9 and the C_9_Mab-24 epitope region.

**Table 1 antibodies-12-00011-t001:** Identification of the C_9_Mab-24 epitope using 1× Ala*-*substituted mCCR9 peptides.

Peptides	Sequences	C_9_Mab-24
WT	MMPTELTSLIPGMFDDFSY	++
M1A	AMPTELTSLIPGMFDDFSY	++
M2A	MAPTELTSLIPGMFDDFSY	++
P3A	MMATELTSLIPGMFDDFSY	++
T4A	MMPAELTSLIPGMFDDFSY	++
E5A	MMPTALTSLIPGMFDDFSY	++
L6A	MMPTEATSLIPGMFDDFSY	++
T7A	MMPTELASLIPGMFDDFSY	++
S8A	MMPTELTALIPGMFDDFSY	++
L9A	MMPTELTSAIPGMFDDFSY	++
I10A	MMPTELTSLAPGMFDDFSY	++
P11A	MMPTELTSLIAGMFDDFSY	++
G12A	MMPTELTSLIPAMFDDFSY	++
M13A	MMPTELTSLIPGAFDDFSY	++
F14A	MMPTELTSLIPGMADDFSY	–
D15A	MMPTELTSLIPGMFADFSY	++
D16A	MMPTELTSLIPGMFDAFSY	++
F17A	MMPTELTSLIPGMFDDASY	–
S18A	MMPTELTSLIPGMFDDFAY	++
Y19A	MMPTELTSLIPGMFDDFSA	++

++, OD655 ≥ 0.3; –, OD655 < 0.1.

**Table 2 antibodies-12-00011-t002:** Identification of the C_9_Mab-24 epitope using 2× Ala*-*substituted mCCR9 peptides.

Peptides	Sequences	C_9_Mab-24
WT	MMPTELTSLIPGMFDDFSY	++
M1A–M2A	AAPTELTSLIPGMFDDFSY	++
M2A–P3A	MAATELTSLIPGMFDDFSY	++
P3A–T4A	MMAAELTSLIPGMFDDFSY	++
T4A–E5A	MMPAALTSLIPGMFDDFSY	++
E5A–L6A	MMPTAATSLIPGMFDDFSY	++
L6A–T7A	MMPTEAASLIPGMFDDFSY	++
T7A–S8A	MMPTELAALIPGMFDDFSY	++
S8A–L9A	MMPTELTAAIPGMFDDFSY	++
L9A–I10A	MMPTELTSAAPGMFDDFSY	++
I10A–P11A	MMPTELTSLAAGMFDDFSY	++
P11A–G12A	MMPTELTSLIAAMFDDFSY	++
G12A–M13A	MMPTELTSLIPAAFDDFSY	++
M13A–F14A	MMPTELTSLIPGAADDFSY	–
F14A–D15A	MMPTELTSLIPGMAADFSY	–
D15A–D16A	MMPTELTSLIPGMFAAFSY	++
D16A–F17A	MMPTELTSLIPGMFDAASY	–
F17A–S18A	MMPTELTSLIPGMFDDAAY	–
S18A–Y19A	MMPTELTSLIPGMFDDFAA	++

++, OD655 ≥ 0.3; –, OD655 < 0.1.

## Data Availability

All related data and methods are presented in this paper. Additional inquiries should be addressed to the corresponding authors.

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
