# Peer review of "Determination of the Binding Epitope of an Anti-Mouse CCR9 Monoclonal Antibody (C9Mab-24) Using the 1× Alanine and 2× Alanine-Substitution Method"

_2073-4468, 2023, doi:10.3390/antib12010011_

Round 1

Reviewer 1 Report

  1. This manuscript describes the determination of the binding epitope of an anti-mouse CCR9 monoclonal antibody (C9Mab-24) using the 1x alanine and 2x alanine-substitution method via enzyme-linked immunosorbent assay. The introduction provides a good background about anti-CCR9 but lack of background about epitope mapping methods. It would be better to provide a background and representative references about using alanine-substitution approach for epitope mapping and justify why this approach (not other methods) was selected for this study. In addition, I am wondering if any epitope mapping of CCR9 were performed in other labs as CCR9 is an attractive target for tumor therapy.
  2. Using the 1x and 2x alanine-substitution methods, critical AA residuals (Phe14 and Phe17) and region (13-MFDDFS-18) involving in C9Mab-24-mCCR9 interaction were identified. I am wondering if the epitope results can be verified using other appropriate methods.
  3. I would like to see the significance of your observations in the discussion section. You may include the potential applications of the observations and tell why your observations are important.
  4. Except the N-terminal region of CCR9, I am wondering if other regions can also involve in CCL25 interactions. You should comment if your approach or any other approach can be used to get a more comprehensive understanding of the CCL25 interactions.

Author Response

1. This manuscript describes the determination of the binding epitope of an anti-mouse CCR9 monoclonal antibody (C9Mab-24) using the 1x alanine and 2x alanine-substitution method via enzyme-linked immunosorbent assay. The introduction provides a good background about anti-CCR9 but lack of background about epitope mapping methods. It would be better to provide a background and representative references about using alanine-substitution approach for epitope mapping and justify why this approach (not other methods) was selected for this study. In addition, I am wondering if any epitope mapping of CCR9 were performed in other labs as CCR9 is an attractive target for tumor therapy.

We added the background about epitope mapping methods with reference in the first paragraph of discussion (line 128-132).

As mentioned in discussion, anti-hCCR9 mAbs (91R and 92R) recognized the N-terminus of hCCR9, the detailed epitope mapping was performed (line 153-156).

However, the critical epitope has not been reported in the case of mAb for CAR-T (ref 21) (line 169).

2. Using the 1x and 2x alanine-substitution methods, critical AA residuals (Phe14 and Phe17) and region (13-MFDDFS-18) involving in C9Mab-24-mCCR9 interaction were identified. I am wondering if the epitope results can be verified using other appropriate methods.

We added the information of C9Mab-24 in the introduction (line 62-64).

Since C9Mab-24 is applicable to flow cytometry, we can perform the alanine-scanning mutagenesis using alanine-substituted mCCR9 overexpressing cells. When a mAb recognizes the conformational epitope (not applicable to ELISA), above strategy is useful.

3. I would like to see the significance of your observations in the discussion section. You may include the potential applications of the observations and tell why your observations are important.

Since we have established several anti-CCR9 mAbs and obtained the information of VH and VC, we expect to apply them to CAR-T therapy. Therefore, we mentioned that in the last sentence of discussion (line 169-171).

4. Except the N-terminal region of CCR9, I am wondering if other regions can also involve in CCL25 interactions. You should comment if your approach or any other approach can be used to get a more comprehensive understanding of the CCL25 interactions.

Currently, the detailed structure of CCR9-CCL25 complex has not been solved. As we cited in the references (3, 25), the mAbs-hCCR9 interaction was partially inhibited in the presence of CCL25. Therefore, other regions of CCR9 can involve in CCL25 interactions. For a comprehensive understanding, we should evaluate both neutralizing and biological activity of our mAbs. We added in discussion (line 158).

Reviewer 2 Report

Kobayashi et al. describe the epitope mapping method and its result using anti-mouse CCR9 mAb, C9Mab-24 (rat IgG2, kappa), which was established by the peptide immunization in their previous work. Although the article is simple, the results clearly show that the Ala scanning method for epitope mapping will be beneficial to estimate the mAb-target peptide interaction. It is versatile and low-cost because the method can be conducted by general ELISA and several synthetic peptides. In this sense, it would be beneficial for antibody researchers to widely disseminate such a useful method in open-access journals.

Minor comments

1. This article seems to argue that 1 x Ala and 2 x Ala scanning using synthetic peptides was effective to elucidate the antibody’s recognition site. Therefore, I think that common epitope mapping methods including Ala scanning, and the merit of their strategy should be mentioned in the introduction or discussion.

2. The authors concluded MFDDFS is a binding site. However, from the 1 x Ala substitution result, it seems M to A slightly affects the ELISA signal and is not essential. If so, the description of M should be added to the discussion part.

Author Response

Kobayashi et al. describe the epitope mapping method and its result using anti-mouse CCR9 mAb, C9Mab-24 (rat IgG2, kappa), which was established by the peptide immunization in their previous work. Although the article is simple, the results clearly show that the Ala scanning method for epitope mapping will be beneficial to estimate the mAb-target peptide interaction. It is versatile and low-cost because the method can be conducted by general ELISA and several synthetic peptides. In this sense, it would be beneficial for antibody researchers to widely disseminate such a useful method in open-access journals.

Minor comments

  1. This article seems to argue that 1 x Ala and 2 x Ala scanning using synthetic peptides was effective to elucidate the antibody’s recognition site. Therefore, I think that common epitope mapping methods including Ala scanning, and the merit of their strategy should be mentioned in the introduction or discussion.

We added the background about epitope mapping methods in the first paragraph of discussion (line 128-132).

  1. The authors concluded MFDDFS is a binding site. However, from the 1 x Ala substitution result, it seems M to A slightly affects the ELISA signal and is not essential. If so, the description of M should be added to the discussion part.

We agree the slight reduction in M13A mutant. However, we could not exclude the possibility of the quality of synthetic peptide.

Therefore, we considered only background levels of reactivity below 0.1 to be negative.

We hope that the reviewer understands the situation.

Reviewer 3 Report

A clear study showing the critical residues required for C9Mab-24 to bind to the N terminal region of mouse CCR9.

Please include an alignment of mouse, rat and human sequences for 1-19 and comment on how Mab-24 was obtained.  Referencing a paper in press (32) for the origin of Mab-24 is not acceptable, and some brief description of how the antibody was raised, implications for breaking tolerance, opportunities for cross-reactivity and potential therapeutic application are needed to strengthen the manuscript.

Author Response

A clear study showing the critical residues required for C9Mab-24 to bind to the N terminal region of mouse CCR9.

Please include an alignment of mouse, rat and human sequences for 1-19 and comment on how Mab-24 was obtained. Referencing a paper in press (32) for the origin of Mab-24 is not acceptable, and some brief description of how the antibody was raised, implications for breaking tolerance, opportunities for cross-reactivity and potential therapeutic application are needed to strengthen the manuscript.

We added the information of C9Mab-24 in the introduction (line 62-64).

We added Figure 1C which shows the alignment. From the alignment and our epitope mapping, C9Mab-24 is expected to recognize mouse CCR9, but not human and rat CCR9.

Reviewer 4 Report

In the manuscript titled “Determination of the Binding Epitope of an Anti-mouse CCR9 Monoclonal Antibody (C9Mab-24) using the 1 × alanine and 2 × alanine-substitution Methodby Hiyori Kobayashi et.al., the authors have identified the epitope of the anti-mCCR9 mAb and using 2 × Ala and 1 × Ala-substitution methods, they found that phenylalanines at 14 and 17 are critical among the amino acids for binding to mCCR9.

 This reviewer would like the authors to address the following points

 (1)   The 1 × Ala-substitution method have shown that phenylalanines at 14 and 17 are critical for binding to mCCR9. This reviewer could not figure out the importance of performing 2 × alanine-substitution method since the M13A-F14A and F14A-D15A are expected to show negative binding result because of the presence of phenylalanine at 14. As this outcome is already expected, then what is the purpose of carrying out the 2 × alanine-substitution method.

(2)   There is a difference in OD655 value between D15A and D16A (in Fig 1. A) and D15A-D16A (in Fig 2.A). This reviewer is curious why double Ala substitution OD655 value is lower than single Ala substitution.

Author Response

In the manuscript titled “Determination of the Binding Epitope of an Anti-mouse CCR9 Monoclonal Antibody (C9Mab-24) using the 1 × alanine and 2 × alanine-substitution Method” by Hiyori Kobayashi et.al., the authors have identified the epitope of the anti-mCCR9 mAb and using 2 × Ala and 1 × Ala-substitution methods, they found that phenylalanines at 14 and 17 are critical among the amino acids for binding to mCCR9.

This reviewer would like the authors to address the following points

(1) The 1 × Ala-substitution method have shown that phenylalanines at 14 and 17 are critical for binding to mCCR9. This reviewer could not figure out the importance of performing 2 × alanine-substitution method since the M13A-F14A and F14A-D15A are expected to show negative binding result because of the presence of phenylalanine at 14. As this outcome is already expected, then what is the purpose of carrying out the 2 × alanine-substitution method.

We added the information of the epitope mapping of Cx6Mab-1. The 2 × Ala-substitution could be another option to determine the epitope of mAbs if the epitope was not determined by the 1 × Ala-substitution method (line 147-152)

(2) There is a difference in OD655 value between D15A and D16A (in Fig 1. A) and D15A-D16A (in Fig 2.A). This reviewer is curious why double Ala substitution OD655 value is lower than single Ala substitution.

It is difficult to compare that because these experiments were not performed at the same time.